# Epithelioid Fibrous Histiocytoma with *CARS-ALK* Fusion: First Case Report

Léo-Paul Secco [1,*], Louis Libbrecht [1,2], Elsa Seijnhaeve [1], Silke Eggers [3], Anne-France Dekairelle [4] and An-Katrien De Roo [1,5]

1    Department of Pathology, Cliniques Universitaires Saint-Luc, B-1200 Brussels, Belgium
2    Department of Pathology, AZ Groeninge, B-1200 Kortrijk, Belgium
3    Department of Dermatology, Clinique Saint-Jean, B-1200 Brussels, Belgium
4    Department of Genetics, Cliniques Universitaires Saint-Luc, B-1200 Brussels, Belgium
5    Institute of Experimental and Clinical Research, UCLouvain, B-1200 Brussels, Belgium
*    Correspondence: leo-paul.secco@saintluc.uclouvain.be

**Abstract:** Epithelioid fibrous histiocytoma (EFH) is a type of uncommon skin tumor mostly harboring Anaplastic Lymphoma Kinase (*ALK*) gene rearrangement, with different fusion partners reported. Whether this tumor is a separate entity or has a relationship with conventional fibrous histiocytomas is still a matter of debate. Benign course is the rule after complete surgical excision. A rare subtype of EFH with fusiform cells has been described, with specific fusion partners. Inflammatory myofibroblastic tumor (IMT) is a type of soft tissue tumor rarer than EFH, and it can display distant metastases. Some cases of primary cutaneous IMT included two with Cysteinyl-tRNA Synthetase 1 (*CARS*)-*ALK* rearrangement. IMT can have the same fusion partners as EFH, such as *DCTN1*, *TMP3* or *EML4* genes. We report the case of a 42-year-old woman presenting EFH with fusiform morphology harboring *CARS-ALK* fusion and discuss similarities and differences with IMT.

**Keywords:** epithelioid fibrous histiocytoma; *CARS-ALK* fusion; inflammatory myofibroblastic tumor; ALK rearrangement

## 1. Introduction

Epithelioid histiocytous fibroma (EFH) is recognized by the World Human Organization classification of skin tumors by a distinct type of fibro-histiocytic proliferation from fibrous histiocytoma, harboring specific clinical and histopathological features. Almost all EFHs have an expression of Anaplastic Lymphoma Kinase (ALK) protein evidenced by immunohistochemistry, which is associated with an *ALK* gene fusion with another partner. Some authors have individualized a particular subtype of spindle cell epithelioid fibrous histiocytoma with specific fusion partners. We report yet another novel fusion partner of *ALK* in EFH, namely the Cysteinyl-tRNA Synthetase 1 (*CARS*) gene. *CARS-ALK* rearrangement was first identified in a metastasis from an inflammatory myofibroblastic tumor (IMT) [1]. Our case report of a novel *CARS-ALK* rearrangement in a spindle cell EFH histologically close to a conventional fibrous histiocytoma raises discussion about the differential diagnosis between fibrous histiocytoma, EFH and IMT (Table 1).

**Table 1.** Comparison chart for the differential diagnosis of fibrous histiocytoma, epithelioid fibrous histiocytoma and inflammatory myofibroblastic tumor.

|  | Fibrous Histiocytoma | Epithelioid Fibrous Histiocytoma | Inflammatory Myofibroblastic Tumor |
|---|---|---|---|
| Frequency in skin | Frequent | Uncommon | Rare |

**Table 1.** *Cont.*

| | Fibrous Histiocytoma | Epithelioid Fibrous Histiocytoma | Inflammatory Myofibroblastic Tumor |
|---|---|---|---|
| Location | Limbs, trunk Head and neck uncommon | Limbs Trunk, head and neck uncommon | Head and neck Soft tissues |
| Histopathology Architecture | Rounded to wedge-shaped dermal-based nodule with epidermal hyperplasia | Nodule with exophytic growth, epidermal collarette | Nodular or multinodular |
| Cellularity | Fibroblastic cells with round to elongated nuclei | Plump epithelioid cells with vesicular nuclei and small nucleoli | Myofibroblastic and fibroblastic spindle cells |
| Stroma | Coarse collagen Macrophages | Numerous small capillaries | Inflammatory infiltrate (plasma cells, lymphocytes) within myxoid or collagenized background |
| Immunohistochemistry | Positivity for CD68 and factor XIIIa, sometimes SMA ALK negative | SMA and desmin negative CD30 or EMA may be expressed ALK positive | CD68 positive in histiocytic-like cells Desmin and SMA variably positive ALK positive |
| Molecular alterations | Non recurrent karyotypic alterations | ALK gene fusions | ALK gene fusions |
| Most common ALK gene fusion partners | | VCL, SQSTM1, EML4, TMP3, PRKAR2A, MLPH, DCTN1, CLTC, PPFIBP1 | TPM3, TPM4, RANBP2, CARS, ATIC LMNA, PRKAR1A, CLTC, FN1, EML4, DCTN1, PPFIBP1 |
| Recurrence or distant metastasis | Rarely | Rarely | Yes |

Abbreviations: SMA: smooth muscle actin; ALK: anaplastic lymphoma kinase; VCL: vinculin; SQSTM1: sequesto-some; TPM3: tropomyosin 3; EML4: echinoderm microtubule-associated protein-like 4; MLPH: melanophilin; PRKAR2A: protein kinase cAMP-dependent type II regulatory subunit alpha; DCTN1: dynactin subunit 1; CLTC: clathrin heavy chain; PPFIBP1: PPFIA Binding Protein 1; TMP4: tropomyosin 4; RANBP2: RAN Binding Protein 2; CARS: Cysteinyl-tRNA Synthetase 1; ATIC: 5-Aminoimidazole-4-Carboxamide Ribonucleotide Formyltrans-ferase/IMP Cyclohydrolase; LMNA: lamin A/C; PRAKR1A: protein kinase cAMP-dependent type II regulatory subunit alpha; FN1: Fibronectin 1.

## 2. Case Report

We have detected a *CARS-ALK* rearrangement in an EFH that concerned a healthy 42-year-old woman presenting with a nodular lesion of the forearm, which clinically resembled a benign fibrous histiocytoma. Histologically, the lesion consisted of a slightly raised, relatively well-circumscribed, unencapsulated dermal nodule, composed of spindled to dendritic cells, arranged in a whorled fashion (Figure 1). Architecture was close to a conventional fibrous histiocytoma (dermal nodule with fibro-histiocytic cells, epidermal hyperplasia). However, cells were plumper than usual, and not associated with a typical coarse collagen at the periphery. We then made the hypothesis of an EFH. Although the tumor cells were not epithelioid, as typically seen in EFH, there was a diffuse cytoplasmic and granular immunoreactivity for the ALK protein, as well as for factor XIIIa and CD68,

but not for smooth muscle actin, consistent with the diagnosis of EFH. The overexpression of the ALK protein correlated nicely with an *ALK* gene rearrangement detected by fluorescence in situ hybridization (using the LSI-Vysis ALK Dual Color Break Apart Rearrangement (Abbott, Chicago, IL, USA.) probe). Upon next generation sequencing, using the FusionPlex® Lung Archer® (Archer, Boulder, CO, USA) panel, the fusion transcript between exon 17 of the *CARS* gene (NM_001751.5, breakpoint chr11:3033425) and exon 20 of the *ALK* gene (NM_004304.4, breakpoint chr2:29446394) was identified. According to this configuration, most of the regions of CARS and the catalytic domain of ALK are retained (Figure 2). The tumor did not recur after ten months of follow-up.

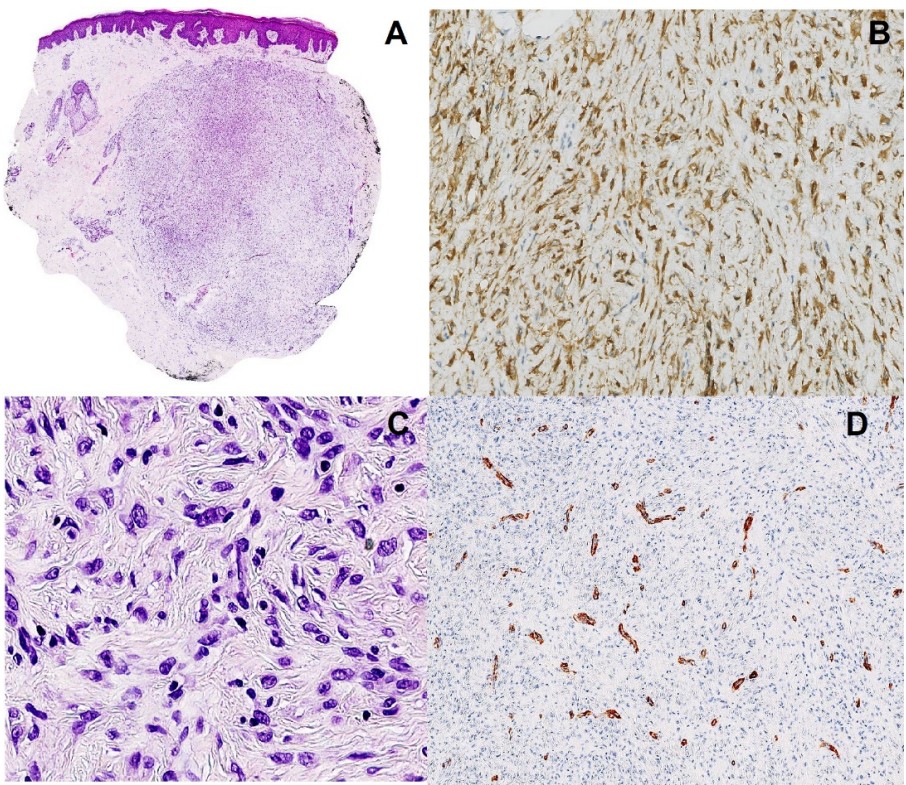

**Figure 1.** Pathological findings. (**A**). Silhouette of the lesion showing epidermal hyperplasia, dermal nodule with endophytic growth (magnification ×12.5). (**B**). Cytoplasmic staining with ALK1 antibody (immunohistochemistry), highlighting the dendritic shape of the cells (magnification ×200). (**C**). Dendritic to epithelioid cells, with ovoid vesicular nuclei and tiny nucleoli, arranged in a whorled fashion (hematoxylin and eosin, magnification ×200). (**D**). Smooth muscle actin immunohistochemistry, showing small vessels within the tumor, without staining of the tumor cells (magnification ×200).

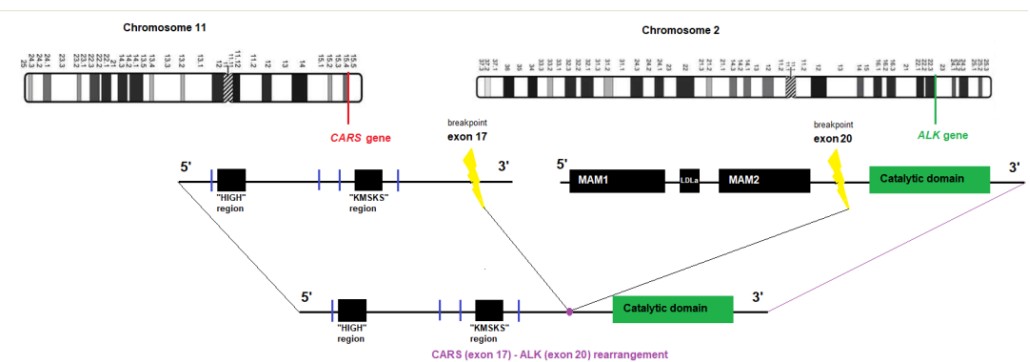

**Figure 2.** Graphical view of the CARS-ALK fusion transcript.

## 3. Discussion and Conclusions

*CARS-ALK* fusions are described in IMT [2], even in skin locations [3,4]. However, to our knowledge, this is the first case to report a *CARS-ALK* fusion in EFH. *CARS* is a gene located in chromosome 11, encoding a class 1 aminoacyl-tRNA synthetase. This gene is one of several located near the imprinted gene domain altered in Beckwith-Wiedemann syndrome, Wilms tumor and other cancers. *ALK* encodes a receptor tyrosine kinase, which belongs to the insulin receptor superfamily, with an intracellular kinase domain. *CARS-ALK* fusion participates in two reactions: ligand-independent dimerization and autophosphorylation of ALK fusion. In IMT, the chimeric fused genes are likely to contribute to the neoplastic transformation by providing an active promoter, leading to overexpression of the ALK fusion product with preserved C-terminal regions, harboring the receptor tyrosine kinase activity and mediating the homo-oligomerization of the chimeric product, leading to activation of the *ALK* gene signaling pathway [5]. By analogy, a similar mechanism might play a role in EFH. The morphological findings in our case closely resemble those of both cases with *CLTC-ALK* fusions described by Georgantzoglou et al., who have identified the *CLTC* gene as a novel fusion partner of the *ALK* gene in two cases of EFH [6]. Indeed, both types of EFH were associated with predominantly fusiform to dendritic cells, instead of epithelioid cells, arranged in a whorled fashion, showing no exophytic growth or epidermal collarette and lacking a prominent capillary component. These morphological features are reminiscent of the spindle cell variant of epithelioid cell histiocytofibroma [7], which has been reported to present *ALK* fusions with *DCTN1*, *TMP3* and *EML4* genes [8]. Furthermore, these fusions have also been identified in IMT [9–11]. IMT and EFH can both show cytoplasmic expression of ALK and factor XIIIa, but, unlike IMT and some classical benign fibrous histiocytomas, EFH does not express smooth muscle actin [12]. Unlike IMT, EFH has no distant metastatic potential. Still, striking similarities can be found between both entities, as IMT can also harbor epithelioid cell morphology and express CD30, as seen in EFH, or may have few inflammatory cells [3,13,14]. Inversely, EFH may show a prominent inflammatory infiltrate, reminiscent of IMT. As molecular pathology can better classify tumors with poor cell differentiation identifying recurrent fusion abnormalities, clinical context and morphology are still important to discriminate tumors with different potential when they share the same fusion genes. Whether these similarities in morphology and molecular pathology represent a true biological relationship between EFH and IMT, defining a spectrum within these two entities, remains subject of future study.

**Author Contributions:** Conceptualization: L.-P.S. and E.S.; methodology: L.-P.S., A.-K.D.R. and A.-F.D.; formal analysis and investigation: L.-P.S., A.-K.D.R. and A.-F.D.; writing—original draft preparation: L.-P.S. and A.-K.D.R.; writing—review and editing: L.-P.S. and A.-K.D.R.; funding acquisition: L.-P.S., A.-K.D.R., E.S. and S.E.; resources: L.-P.S., A.-K.D.R. and L.L.; supervision: L.-P.S., A.-K.D.R. and L.L. All authors have read and agreed to the published version of the manuscript.

**Funding:** This research received no external funding.

**Institutional Review Board Statement:** Not applicable.

**Informed Consent Statement:** Not applicable.

**Data Availability Statement:** No new data were created or analyzed in this study. Data sharing is not applicable to this article.

**Acknowledgments:** We acknowledge Jonathan Vanderveken for his help.

**Conflicts of Interest:** The authors have no relevant financial or non-financial interests to disclose.

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
