# Peer review of "Epithelioid Fibrous Histiocytoma with CARS-ALK Fusion: First Case Report"

_dermatopathology, doi:10.3390/dermatopathology10010003_

Round 1

Reviewer 1 Report

The reviewer wishes to thank the editor and the authors for the opportunity to review this concise manuscript regarding a novel CARS-ALK fusion in an epithelioid fibrous histiocytoma (EFH).  The manuscript identified a case of EFH with a spindled to dendritic morphology that was positive for ALK immunohistochemical staining and on NGS was found to have a CARS-ALK fusion.

The manuscript is exceptionally interesting given this novel fusion and the possible comparison with IMT.  The manuscript may be more attractive to readers if the body of the manuscript is split into various sections including introduction, case report, discussion and conclusion.

Additionally, further description of function and definition of each fusion partner may help readers better understand possible implications of these translocations in the formation of these lesions. Additionally, further clarification of the classic histomorphology of the fibrous histiocytoma, EFH, and IMT may be of help (perhaps a comparison chart could be considered).

While the ALK IHC image in Figure 1 is beautiful, Figure 1C appears pale and Figure 1D is hard to interpret and may benefit from a higher power image.

Author Response

Point 1: The manuscript may be more attractive to readers if the body of the manuscript is split into various sections including introduction, case report, discussion and conclusion.

Response 1:  The manuscript have been splitted in several parts.

Point 2: Additionally, further description of function and definition of each fusion partner may help readers better understand possible implications of these translocations in the formation of these lesions.

Response 2: A graphical view have been added as suggested by other reviewer and explanations of the pathophysiology are developed lines 59-69.

Point 3: Additionally, further clarification of the classic morphology of the fibrous histiocytoma, EFH, and IMT may be of help (perhaps a comparison chart could be considered).

Response 3 : Keypoints of the differential diagnosis about conventional fibrous histiocytoma, epithelioid fibrous histiocytoma and inflammatory myofibroblastic tumor have been reported in a comparison chart (Table 1).

Point 4: While the ALK IHC image in Figure 1 is beautiful, Figure 1C appears pale and Figure 1D is hard to interpret and may benefit from a higher power image.

Response 4: We modified Figure 1 with a higher HE to better visualize nuclei of the cells, remove FISH because our camera cannot provide beautiful images of the slide, and added smooth muscle actin immunohistochemistry, as it is crucial for differential diagnosis with IMT.

Reviewer 2 Report

This paper is interesting in that it finds a novel CARS-ALK fusion in epithelioid fibrous histiocytoma (EFH). However, several aspects of the data are not convincing.

1) As the authors describe, differentiating EFH from IMT is problematic; since negative staining for αSMA is a hallmark of EFH, please show that staining.

2) A schematic diagram of the CARS-ALK fusion gene product would help the reader better understand.

3) The order of panel A-B-C in Figure 1 should be modified to C-A-B.

4) The resolution of the FISH in panel D is very poor, and the red and green dots are not clear. Also, there are no yellow dots indicating fusion proteins.

5) The resolution of panel B does not clearly show ovoid vesicular nuclei and tiny nucleoli. 

6) What does CARS stand for?

7) The magnification in panel C is not shown.

8) Line 58: “unlike” is duplicated. 

9) Line 71: change nucleous to nuceoli.

Author Response

Thank you for your comments. Here are the modifications of the article you suggested.

Point 1: As the authors describe, differentiating EFH from IMT is problematic; since negative staining for αSMA is a hallmark of EFH, please show that staining.

Response 1: alpha SMA staining is shown in Figure 1D.

Point 2: A schematic diagram of the CARS-ALK fusion gene product would help the reader better understand.

Response 2: A graphical view of the fusion transcript has been added in Figure 2.

Point 3: The order of panel A-B-C in Figure 1 should be modified to C-A-B.

Response 3: The order have been modified.

Point 4: The resolution of the FISH in panel D is very poor, and the red and green dots are not clear. Also, there are no yellow dots indicating fusion proteins.

Response 4: We agree with that, unfortunately, our camera cannot provide better quality image from FISH, we then decided to remove it.

Point 5: The resolution of panel B does not clearly show ovoid vesicular nuclei and tiny nucleoli.

Response 5: The magnification of HE has been raised at x400 to better visualize the nuclei of the cells.

Point 6: What does CARS stand for?

Response 6: More explanations of this gene are now available lines 59 to 69.

Point 7: The magnification in panel C is not shown.

Response 7: Now in panel A, magnification x 12,5

Point 8: Line 58: “unlike” is duplicated. 

Response 8: Corrected.

Point 9: Line 71: change nucleous to nuceoli.

Response 9: Corrected.

Round 2

Reviewer 2 Report

The authors have adequately responded to my comments; it is unfortunate that they were unable to include a clear image of FISH, but instead the fusion gene is clearly illustrated in Figure 2. The revised manuscript is much improved than the original and is easier to understand.